# Second Cancer Onset in Myeloproliferative Neoplasms: What, When, Why?

**DOI:** 10.3390/ijms23063177

**Published:** 2022-03-15

**Authors:** Cosimo Cumbo, Luisa Anelli, Antonella Zagaria, Nicoletta Coccaro, Francesco Tarantini, Giorgina Specchia, Pellegrino Musto, Francesco Albano

**Affiliations:** 1Hematology and Stem Cell Transplantation Unit, Department of Emergency and Organ Transplantation (D.E.T.O.), University of Bari “Aldo Moro”, 70124 Bari, Italy; cosimo.cumbo@uniba.it (C.C.); luisa.anelli@uniba.it (L.A.); antonellazagaria@hotmail.com (A.Z.); nicoletta.coccaro@uniba.it (N.C.); francescotarantini14@gmail.com (F.T.); pellegrino.musto@uniba.it (P.M.); 2School of Medicine, University of Bari “Aldo Moro”, 70124 Bari, Italy; specchiagiorgina@gmail.com

**Keywords:** myeloproliferative neoplasms, second cancers, solid tumors

## Abstract

The risk of developing a solid cancer is a major issue arising in the disease course of a myeloproliferative neoplasm (MPN). Although the connection between the two diseases has been widely described, the backstage of this complex scenario has still to be explored. Several cellular and molecular mechanisms have been suggested to link the two tumors. Sometimes the MPN is considered to trigger a second cancer but at other times both diseases seem to depend on the same source. Increasing knowledge in recent years has revealed emerging pathways, supporting older, more consolidated theories, but there are still many unresolved issues. Our work aims to present the biological face of the complex clinical scenario in MPN patients developing a second cancer, focusing on the main cellular and molecular pathways linking the two diseases.

## 1. Introduction

Myeloproliferative neoplasms (MPNs) (polycythemia vera (PV), essential thrombocythemia (ET), primary myelofibrosis (PMF), chronic myeloid leukemia (CML), and unspecified MPNs) are chronic hematological cancers featuring different progression rates [1]. Although considered to be relatively indolent malignancies, patients suffer an increased risk of vascular complications, disease progression, leukemic transformation, and hence a reduced life expectancy [1,2].

In recent years, a wide range of studies has shown an increased risk of developing a second cancer (SC) in the different MPN subtypes as compared to the general population [3,4,5,6,7].

A recent meta-analysis collecting data on more than 65,000 MPNs from 12 large studies summarized the main epidemiologic, prognostic and clinical aspects of this condition, aiming to provide clinicians with an overview supporting the disease management [8]. The biological background underlying this complex scenario has perhaps been less widely studied but is clearly equally important. Some biological aspects connecting the two events are MPNs-dependent, others depend on treatment, and still others are bridge mechanisms between the two; often, several processes overlap [8].

The presented work aims to collect data on the main cellular and molecular pathways linking the two neoplasms occurring in MPN patients who develop an SC (Table 1), focusing on what happens and trying to understand when and why.

## 2. Risk of SC Onset in MPNs: An Overview

The frequency of a second solid or lymphoid cancer is increased in patients with chronic MPNs as compared with age-/sex-matched healthy individuals, with the risk of SC such as lymphomas and tumors of the skin, lung, kidney, and thyroid gland being 1.5–3.0-fold higher in MPN patients, especially in the age group spanning 60–80 years [8]. Conversely, the risk of other solid neoplasms such as colon, breast, and prostate cancer was not different from that in the rest of the population [8]. Several large studies followed patients with varying subtypes of MPN and revealed a cumulative incidence of SC ranging from 5 to 10% after 5 years from diagnosis. The different subtypes of MPNs have similar relative increases in the risk of an SC, but the time of onset is shorter in PMF as compared to PV and ET [4].

Few data are available about the prognosis of cancer patients with previous MPNs. A single work studied this aspect, comparing 1246 MPNs and one SC with 5155 age/sex-matched patients with the same cancers but without preceding MPNs [38]. The study showed that survival was significantly poorer for cancer patients following MPN than other cancers, the hazard ratio for death being increased 1.5-fold for cancer patients with antecedent ET, 1.2-fold with PV, and 1.2-fold with CML [38]. Cardiovascular events, thrombosis and infections are the main causes of death in patients with MPNs and SC, and managed by supportive cancer care, such as anti-aggregant and antibiotic therapy [38]. In any case, a preceding MPN can be considered a predictor for poor outcomes in patients who develop new primary cancers [38]. Regarding the management of these patients, extra surveillance measures during follow-up may be considered in patients aged 60–79 years; systematic skin inspection and imaging analyses may be recommended, since a large proportion of SC are skin, kidney and lung cancers [38].

## 3. Possible Links between MPNs and SC

The pathogenic mechanisms responsible for the increased SC risk in MPNs have not been elucidated. Different hypotheses suggest the presence of shared genetic risk factors and an inherent tendency to develop cancer, or the effects of cytotoxic anti-neoplastic treatment, or else a possible link with chronic inflammation or immune dysfunction. All these aspects will be considered below.

### 3.1. Genetic Susceptibility

One possible molecular mechanism at the basis of the increased SC frequency in MPNs is a hereditary susceptibility to developing cancer that could confer an intrinsic predisposition even in untreated MPN patients [39]. Although somatic driver mutations cause the onset of MPNs in *JAK2*, *CALR*, and *MPL* genes or the *BCR/ABL1* fusion gene in CML, several recent epidemiological studies revealed a crucial heritable component of the disease. In fact, first-degree relatives of patients with MPNs show a 5–7-fold increased risk of developing MPN compared to that observed in solid familial cancers like breast, prostate, and colorectal cancer [39,40]. Several genome-wide association (GWA) studies in MPN family clusters have identified many germline genetic variants associated with an increased risk of developing MPN. The strongest germline risk factors identified so far are the presence of the *JAK2* 46/1 haplotype and telomerase reverse transcriptase (*TERT*) gene variants (Table 2), although these account for only a small fraction of familial MPN (Figure 1) [41]. The *JAK2* 46/1 haplotype, also referred to as “GGCC”, spans the *JAK2* gene and is a “pre-*JAK2*” event predisposing to the acquisition of the *JAK2 V617F* mutation [42,43]. This haplotype accounted for a large proportion of sporadic MPNs, but does not account for a family predisposition, as this is not shared by all family members [44].

It is well known that abnormal telomerase activity plays a crucial role in the development of several cancers, and the *TERT* gene encodes a reverse transcriptase of the telomerase complex [9]. Mutations in the *TERT* gene can alter telomerase activity and telomere length, inducing bone marrow failure syndromes and significantly increasing cancer frequency [10]. Moreover, several SNPs located in exon or intron regions seem to influence telomere length and have been associated with the risk of different cancer types. The two most commonly studied SNP variants of the *TERT* gene are rs2736100 and rs2736098, which are related to the risk of both hematologic and solid cancers: rs2736100 was associated with an increased risk of thyroid, bladder, and lung cancer, glioma, and MPNs, whereas rs2736098 increased the risk of bladder and lung cancers. The *TERT* rs2736100 variant represents a germline predisposing factor with a non-specific effect on all MPNs, regardless of phenotype (PV, ET or PMF) or major molecular subtype such as the occurrence of *JAK2* V617F or *CALR* gene mutations [10,45].

In several studies of GWAS, whole-exome sequencing (WES) or SNP arrays performed in MPN patients identified germline mutations or common genetic polymorphisms in multiple genes including *MECOM*, *HBS1L*-*MYB*, *SH2B3* (*LNK*), *TET2*, *ATM*, *CHEK2*, *LINC*-*PINT*, *ATG2B*, *GSKIP*, *RBBP6* and *GFI1B*, or epigenetic modifiers as *ASXL1*, *DNMT3A*, *TET2*, *IDH2*, and *NF1* (Figure 1) [11,12,39,40,46]. These genes are involved in crucial cellular pathways regulating cell proliferation, differentiation or apoptosis and are also associated with a predisposition to several solid cancers [47,48,49].

Other genes that can predispose to the onset of either MPNs or solid cancers are included in the new WHO 2016 category of “Myeloid neoplasms with germline predisposition”, representing a rare but underdiagnosed entity whose recognition is recognized as critical for proper patient clinical management [50]. Patients and their family members should be closely monitored because they show an increased risk of non-hematopoietic malignancies and other organ dysfunctions. Myeloid neoplasms with germline predisposition are most frequently acute myeloid leukemia (AML) and myelodysplastic syndromes (MDS), more rarely other neoplasms such as CML, atypical CML (aCML), chronic myelomonocytic leukemia (CMML) or PV and PMF. Mutated genes include *CEBPA*, *DDX41*, *RUNX1*, *ANKRD26*, *ETV6*, and *GATA2*, which are inherited in autosomal dominant patterns (Figure 1) [50,51].

The *DDX41* gene encodes an RNA helicase protein involved in RNA splicing (Table 2) [52,53]. Multiple germline variants have been described, including frameshift, missense, and splicing mutations. Most patients with *DDX41* mutations present an AML or MDS, but some cases are affected by CML or other MPNs, and some families also have a predisposition to immune disorders such as lupus, eczema, or vasculitis [52,53]. Some patients with a myeloid neoplasm and *DDX41* germline mutations have a family history of solid cancers in first- or second-degree relatives or personal records of solid cancers, such as renal cell, prostate, and breast carcinoma [53].

The *ETV6* missense mutations have a dominant-negative effect, resulting in a disrupted nuclear localization of the ETV6 transcription factor (Table 2) and the reduced expression of platelet-associated genes [54]. Families with germline *ETV6* mutations most frequently developed B lymphoblastic leukemia/lymphoma, MDS/AML, CMML, multiple myeloma, and PV [54]. A familial predisposition to solid tumors, including colorectal, breast, kidney, and skin cancer and meningioma was also observed [54].

The *GATA2* gene encodes a zinc-finger transcription factor that is considered a master regulator of early hematopoiesis, as it plays a crucial role in the proliferation and maintenance of hematopoietic stem cells (HSCs) (Table 2) [55]. *GATA2* expression is not limited to hematopoietic cells, but is also detected in endothelial, fetal liver and heart, placenta, and central nervous system cells [55]. *GATA2* mutations can be either inherited or acquired, and due to the crucial role that *GATA2* plays in the development and function of several cell lineages, almost all carriers of the mutation will probably develop hematologic or immunologic defects during their lifetime. *GATA2* germline mutations have been reported in familial MDS and AML cases presenting at a younger age, and are also observed in individuals showing the “MonoMAC or DCML deficiency”, an immunodeficiency condition involving monocytes, CD4+ cells, dendritic cells, B and NK lymphoid cells and an increased risk of developing myeloid leukemia and mycobacterial, human papillomavirus (HPV) and opportunistic fungal infections [56]. *GATA2* germline mutations are also detected in patients with “Emberger syndrome”, which is characterized by congenital deafness and lymphedema [56]. In individuals with *GATA2* germline mutations, the onset of a malignant disease is common, frequently due to HPV-driven intraepithelial neoplasia but an increase in breast cancer, squamous cell carcinoma, and neoplasms correlated with Epstein–Barr virus (EBV) infection has also been reported [56]. A recent report described a patient with a *JAK2V617F*-positive PMF and an inherited *GATA2* mutation who developed a basal cell carcinoma of the facial skin six years after the initial diagnosis and therapy with the JAK inhibitor ruxolitinib [57]. The patient showed the germline *GATA2*-N317S missense mutation previously reported in MDS and AML patients but not in cases with GATA2-deficiency syndrome [57]. Patients with a germline *GATA2* mutation are known to be at increased risk for skin cancers such as basal cell carcinoma [57]. In this patient, the *GATA2* mutation preceded the acquisition of the *JAK2V617F* mutation causing the PMF onset [57].

Moreover, it has been hypothesized that CML patients have an inborn increased predisposition to develop secondary malignancies [58]. CML patients show a high risk of developing an SC such as gastrointestinal and nose and throat tumors. However, it has been hypothesized that the augmented incidence may be linked to CML itself, as the prevalence of malignancies and autoimmune diseases is increased in CML patients before their CML diagnosis [58]. These data suggest that a hereditary or acquired predisposition to cancer and immune dysfunction could be involved in the CML pathogenesis.

### 3.2. Effect of Cytotoxic Drugs

Undoubtedly, one of the most evident links between MPNs and SC is the cytotoxic effect of treatment [4,5,13]. In 2019, an international nested case–control study (involving 30 centers) was conducted, aiming to evaluate the risk of SC after exposure to cytoreductive drugs, in a cohort of 1881 Philadelphia-negative MPNs [4]. The administration of hydroxyurea (HU), pipobroman, ruxolitinib and their combination showed an increased risk of non-melanoma skin cancers onset compared with unexposed patients [4]. Notably, very recent studies showed the higher incidence of SC in the post-ruxolitinib era [59], in both PV and myelofibrosis patients [60,61]. Conversely, no association with the risk of overall SC was observed after exposure to interferon-alpha (IFN-alpha), busulfan and anagrelide [4]. Regarding the probability of leukemic conversion, chlorambucil, phosphorus-32 (32P) and pipobroman were the MPN therapies showing the highest risk [62].

Among the drugs mentioned above, HU is the most frequently used cytoreductive agent (due to its perceived efficacy and tolerability), and its mode of action is worthy of discussion [62]. HU is an anti-metabolite agent inhibiting the enzymatic activity of ribonucleotide reductase, thereby inactivating DNA synthesis [62]. HU inhibits DNA synthesis and DNA repair (Table 1). In fact, it increases the number of DNA breaks, causing the strands to remain open longer, and decreases the DNA polymerase activity, thereby slowing the polymerization rate at the repair sites [13,14]. Furthermore, HU and exposure to UV radiation play a combined role in skin cancers onset; in fact, UV-B rays promote the proliferation of *TP53* mutant keratinocytes (seen in the dermo-epidermal junction and hair follicles) and enable them to colonize the adjacent compartment [15,16]. In the basal layer of the epidermis, the high keratinocyte turnover with HU impaired DNA synthesis and repair are the causes of skin tumors onset [61]. For these reasons, during HU administration, patients should avoid excessive sunlight exposure and use chemopreventive agents like oral retinoids [63,64].

Moreover, HU administration was shown to be linked to the occurrence of *PPM1D* truncating mutations. Together with *TP53*, *PPM1D* is another DNA repair gene whose variants onset is widely associated with prior chemotherapy exposure (Table 1) [17,18].

As regards drugs cytotoxicity, use of tyrosine kinase inhibitors (TKI) in chronic myeloid leukemia (CML) is noteworthy. TKI administration is frequently prolonged for decades, and the risk of SC is higher in the CML than the expected rates in the rest of the population [65,66]. Nevertheless, these neoplasms are unlikely to manifest until after several years of treatment [65]. On the contrary, the SC standardized incidence ratio was the same before and two years after diagnosis, suggesting that it is not TKI treatment that causes the increased number of SC, but rather the CML disease itself [65,67]. Therefore, there is no current evidence supporting a link between TKI exposure and the risk of developing SC.

### 3.3. Influence of Chronic Inflammation

Several decades ago, Virchow established a link between inflammation and malignancies, and many cellular and molecular circuits have been described over the years [68,69,70]. MPNs are characterized by a state of chronic inflammation (CI), which is proposed to be the common denominator for premature atherosclerosis onset, clonal evolution, and SC development [30]. MPNs can be considered as a “Human Inflammation Model” since the disease per se produces a state of CI due to the continuous release of inflammatory molecules from activated leukocytes and platelets [19,30,71]. Increasing release of cytokines, chemokines and reactive oxygen species (ROS) gives rise to genetic and epigenetic changes, inducing genomic instability, which thereby contributes to tumor initiation (Table 1) [19]. IL-6, IL-1b, TNF-alpha and ROS are the main factors inducing DNA methylation changes [23,24,25]. Moreover, oxidative DNA damage implies an increased risk of mutagenesis [20,21]. Furthermore, inflammatory mediators activate transcription factors such as NF-KB and STAT3, associated with an altered expression of several genes and then playing major roles in linking inflammation and carcinogenesis (Table 1) [22,72].

In recent years, the ability of inflammation to induce clonal hematopoiesis (CH) has been amply demonstrated [26]. In particular, in vitro, *Tet2*-deficient murine and *TET2*-mutant human hematopoietic stem cells have a strong proliferative advantage compared with wild-type cells when exposed to high levels of exogenous, pro-inflammatory IL-6 and TNF-α (Table 1) [27,28]. CH is associated with an approximately tenfold increased risk (or higher risk with larger clones) of future hematological cancers, mainly MDS and AML [73,74]. Furthermore, CH is now understood to be a risk factor for therapy-related myeloid neoplasms [75]. Among the main CH-related genes, *DNMT3A* and *TET2* have a clear functional role in supporting cancer development [76,77]. It is, therefore, reasonable to suppose that in the inflammatory context of MPNs, CH onset could pave the way for a second hematological cancer.

Lastly, the inflammatory milieu may induce immune deregulation and then a defective tumor immune surveillance, favoring the occurrence of an SC, but this scenario warrants separate discussion [78,79,80,81].

### 3.4. Immune Deregulation

The defective tumor immune surveillance observed in MPNs is a bridge mechanism between the above-described aspects; in fact, all the scenarios presented can, in different ways, affect the immune system ability to conduct surveillance and sense the onset of an SC in MPN patients.

The off-target immunological effect of TKI is an emblematic example: several in vitro studies and animal models show how imatinib can affect the function and differentiation of antigen-presenting cells (APCs) and inhibit the effector functions of T lymphocytes [78,79]. Moreover, the induction of specific cytotoxic T cells seems to be impaired in CML patients treated with imatinib compared with patients receiving IFN-alpha [78]. Another representative example is the NF-KB pathway, which has a key role in inflammation and innate immunity and promotes tumor development [80].

Nevertheless, in MPNs, regardless of drug administration or CI, the release of immunosuppressive cytokines such as VEGF and TGF-beta may be critical for the increased risk of SC development [29]. In fact, both cytokines induced qualitative and quantitative alterations in immune cells that are essential for tumor immune surveillance (e.g., dendritic cells (DCs), cytotoxic T cells, regulatory T cells (T-reg), and natural killer (NK) cells) (Table 1) [30,31].

VEGF mediates the induction and maintenance of CD4(+)CD25 (high) T cells (T-reg), immune cells with a high immunosuppressive power [32]. TGF-beta affects T lymphocytes and APCs, reducing the IL-2–dependent proliferation of T cells by blocking IL-2 production, inhibiting the maturation of T cells and preventing naive T lymphocytes from acquiring effector functions [33,34]. TGF-beta also has potent effects on professional APCs, inhibiting tissue macrophage activation and promoting DCs differentiation from precursors [33,35,36].

Last but not least, signs of immune deregulation and CI were also observed in the gene expression profile of MPNs. In fact, 123 differentially expressed genes involved in these mechanisms were identified in ET, PV and PMF peripheral blood cells, such as *IL*-*4*, *ITGB3*, *TNFAIP8L1*, *SELPLG*, *CREB1*, *LSP1*, *IL1A*, *FAS* (Table 1) [37].

These observations support the use of immune-enhancing therapy in MPNs, aimed at restoring the defective tumor immune surveillance system [29]. IFN-alpha shows this ability to act on crucial immune cells (T cells, DCs and NK cells) involved in these processes and may potentially reduce the increased risk of SC in MPNs [81].

## 4. Conclusions

From the hematologist’s point of view, SC onset in MPNs is one of the risks the patient can run during follow-up, whose occurrence affects the clinical course of the disease. On the contrary, from the oncologist’s point of view, a previous MPN is considered predictive poor outcomes in patients who develop a new primary cancer. In both cases, the patient needs dedicated management considering the co-occurrence of two neoplasms in the same individual.

Regardless of the direction from which the scenario is observed, the mechanisms linking the two events are the same and more care and investigation needs to be devoted to better elucidating them.

In fact, not all MPNs without distinction show the same risk of developing an SC; not all genetic backgrounds predispose equally to these events and not all therapeutic approaches increase the risk in the same way. These considerations underline the need for a personalized evaluation of each case, searching for the most likely molecular (or cellular) pathways linking the two neoplasms.

A more comprehensive biological knowledge of this fascinating and incomplete picture is necessary, particularly probing emerging aspects such as CH onset and potential. New perspectives could emerge from this widely studied topic but even offering open questions.

## Figures and Tables

**Figure 1 ijms-23-03177-f001:**
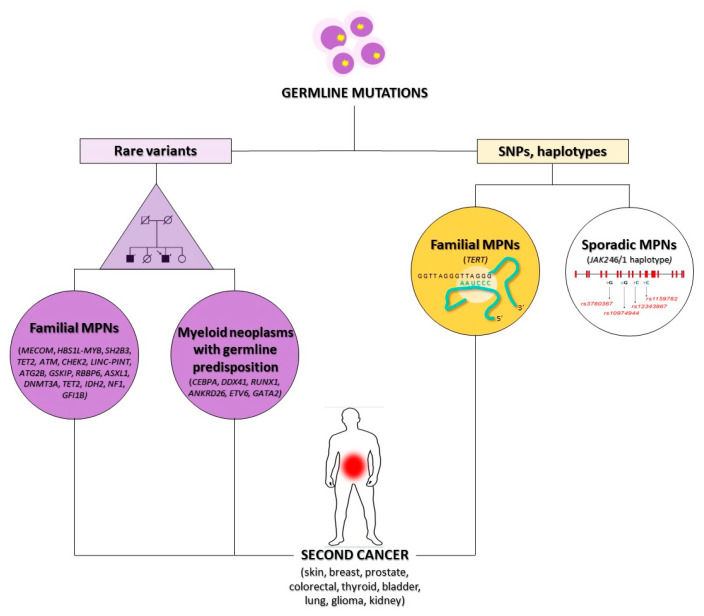
Genetic susceptibility to SC onset in MPNs. The occurrence of rare (on the left) or polymorphic (on the right) germline mutations is one of the possible mechanisms connecting MPNs to an SC onset. SC: second cancer, MPNs: myeloproliferative neoplasms, SNPs: single nucleotide polymorphisms.

**Table 1 ijms-23-03177-t001:** Main cellular and molecular pathways linking the two neoplasms in MPN patients who develop an SC.

Mechanism	Main Pathways	References
Genetic susceptibility	- Telomerase activity (*TERT*) - Epigenetic modification - Cell cycle regulation, differentiation, apoptosis	[9,10,11,12]
Cytotoxic drugs effect	- DNA repair inhibition (*TP53* and *PPM1D* mutations)	[13,14,15,16,17,18]
Chronic inflammation influence	- Cytokines, chemokines and ROS release - NF-KB and STAT3 activation - Clonal hematopoiesis induction	[19,20,21,22,23,24,25] [22] [26,27,28]
Immune deregulation	- APCs, T cells and NK cells suppression (VEGF and TGF-beta release) - Genes deregulation (*IL-4*, *ITGB3*, *TNFAIP8L1*, *SELPLG*, *CREB1*, *LSP1*, *IL1A*, *FAS)*	[29,30,31,32,33,34,35,36] [37]

ROS: reactive oxygen species, APCs: antigen presenting cells, NK: natural killer.

**Table 2 ijms-23-03177-t002:** Main genes involved in the genetic susceptibility to SC onset in MPNs.

Gene Name	Encoding Protein Name	Function
*JAK2*	Homo sapiens Janus kinase 2	Protein tyrosine kinase involved in a specific subset of cytokine receptor signaling pathways
*TERT*	Homo sapiens telomerase reverse transcriptase	Ribonucleoprotein polymerase that maintains telomere ends by addition of the telomere repeat TTAGGG
*DDX41*	Homo sapiens DEAD (Asp-Glu-Ala-Asp) box polypeptide 41	Putative RNA helicases characterized by the conserved motif Asp-Glu-Ala-Asp (DEAD) and involved in RNA splicing
*ETV6*	Homo sapiens ETS variant 6	ETS family transcription factor required for hematopoiesis and maintenance of the developing vascular network
*GATA2*	Homo sapiens GATA binding protein 2	Zinc-finger transcription factor regulating genes involved in the development and proliferation of hematopoietic and endocrine cell lineages

SC: second cancer, MPNs: myeloproliferative neoplasms.

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
