# Peer review of "Second Cancer Onset in Myeloproliferative Neoplasms: What, When, Why?"

_ijms, 2022, doi:10.3390/ijms23063177_

Round 1

Reviewer 1 Report

t is known that patients with chronic MPN have an increased risk of developing acute leukemia, but in recent years it has been shown that there is an increased risk of developing solid cancers. The authors of this paper provide an overview of different scenarios that can lead to the development of secondary cancer, with an emphasis on the cellular and molecular pathways that connect these diseases. It would be desirable for the authors to cite the results of their research in this review and refer to their references.

Author Response

Dear Reviewer,
thanks a lot for your helpful suggestions.
Clinical management and molecular evaluation of MPN patients are our main topics of interest. We have been interested in the onset of second cancer for a long time now. We have recently planned new research projects mainly focusing on studying genetic susceptibility, the first possible mechanism we reported in our manuscript.
In previous studies, we have examined the role of the JAK2 46/1 haplotype in the predisposition to develop MPN.  Furthermore, in the last years, we have been studying the role of chronic inflammation in MPN pathogenesis. As suggested, the relative references (42, 43, 73) have been added.

Reviewer 2 Report

The manuscript is concise, written in good English, and exquisitely organized. However, it needs some minor fine-tuning on a few points.
1. The list of references needs revision: only a few 2021 publications (only 4 out of 75).
2. The phrase "The issue will not just go away!" (line 289) is too enthusiastic and needs to be revised.

Author Response

Dear Reviewer,

thanks a lot for your helpful suggestions.

Some other recent publications have been included (Reff. 60-62, lines 189-190) and the concluding phrase has been revised (lines 299-300), as recommended.

Reviewer 3 Report

Dear Editor,

Thanks for inviting me to evaluate the article titled “Second Cancer Onset in Myeloproliferative Neoplasms: What, When, Why?” In this paper, the authors reviewed the possible connections and the current evidence of the patients with Myeloproliferative neoplasms (MPNs) to develop a second cancer (SC).  The review is well stated but still need some modifications.

One aspect that need to be improved is that there are so many arguments in this review article lack the original sources, which means more references need to be added. The following places are where I can find that need more references:

  1. In line 40, after the words “several processes overlap”;
  2. In line 53, after the words “the rest of the population”;
  3. In line 65, after the words “antibiotic therapy”;
  4. In line 78, after the words “new primary cancers”;
  5. In line 73, after the words “been elucidated”;
  6. In line 80, after the words “untreated MPN patients”;
  7. In line 83, before the words “In fact”;
  8. In line 98, after the words “frequency”;
  9. In line 128, after the words “RNA splicing”;
  10. In line 130, before the words “Most patients”;
  11. In line 138, after the words “platelet-associated genes”;
  12. In line 144, after the words “hematopoietic stem cells (HSCs)”;
  13. In line 154, after the words “fungal infections”;
  14. In line 157, after the words “lymphedema”;
  15. In line 164, after the words “deficiency syndrome”;
  16. In line 165, after the words “basal cell carcinoma”;
  17. In line 196, after the words “skin tumors onset”;
  18. In line 238, after the words “separate discussion”.

Besides the places I listed above, there may be more places need more reference (s), the authors should check the article carefully to ensure every evidence they presented in this article are trackable.

Also in this article, there are some format errors and incorrect used words that need to be changed.

  1. In line 138, after the words “germline ETV6 mutations”, there are more than one space;
  2. In line 194, after the words “adjacent compartment”, there are more than one space;
  3. In line 227, after the words “hematopoietic stem cells (HSCs) have” there are more than one space;
  4. In line 234, before the words “CH onset”, there are more than one space;
  5. In line 269, before the words “restoring the defective” there are more than one space;

  1. In line 41, the words “The present work” need to be changed to “The presented work”;
  2. In line 202, the words “(TKI) use” need to be changed to “(TKI) used”;
  3. In line 215, the words “could be considered a” need to be changed to “could be considered as a”;
  4. In line 253, the words “both cytokines induce” need to be changed to “both cytokines induced”;
  5. From line 119 to line 122, starting with “Myeloid neoplasms...” and ending with “…PV and PMF”, I couldn’t really understand this sentence.

Author Response

Dear Reviewer,

thanks a lot for your helpful suggestions.

All arguments you listed (1-18), derive from references reported before/later in the text. The appropriate references have been added in the lines suggested.

All errors listed (19-28) have been corrected.

Reviewer 4 Report

The manuscript by Cumbo C. et al. summarizes the current knowledge about second cancer onset in myeloproliferative neoplasms.

It would be helpful to show a list with the abbreviations used.

In addition, a table with the gene and encoding protein name with a short description of their function would make it easier to understand the pathogenesis.

In Fig. 1 (second cancer), the authors should clarify what is meant with renal cell and kidney.

Author Response

Dear Reviewer,

thanks a lot for your helpful suggestions.

The list of abbreviations and the table requested, have been added.

Figure 1 has been modified.